# On the Formation of Nanocrystalline Grains in Metallic Glasses by Means of In-Situ Nuclear Forward Scattering of Synchrotron Radiation

**DOI:** 10.3390/nano9040544

**Published:** 2019-04-04

**Authors:** David Smrčka, Vít Procházka, Vlastimil Vrba, Marcel B. Miglierini

**Affiliations:** 1Department of Experimental Physics, Faculty of Science, Palacký University Olomouc, 17. listopadu 12, 771 46 Olomouc, Czech Republic; david.smrcka@upol.cz (D.S.); vlastimil.vrba@upol.cz (V.V.); 2Institute of Nuclear and Physical Engineering, Faculty of Electrical Engineering and Information Technology, Slovak University of Technology in Bratislava, Ilkovičova 3, 812 19 Bratislava, Slovakia; marcel.miglierini@stuba.sk

**Keywords:** nuclear forward scattering, metallic glasses, magnetic annealing, synchrotron radiation, crystallization kinetics

## Abstract

Application of the so-called nuclear forward scattering (NFS) of synchrotron radiation is presented for the study of crystallization of metallic glasses. In this process, nanocrystalline alloys are formed. Using NFS, the transformation process can be directly observed during in-situ temperature experiments not only from the structural point of view, i.e., formation of nanocrystalline grains, but one can also observe evolution of the corresponding hyperfine interactions. In doing so, we have revealed the influence of external magnetic field on the crystallization process. The applied magnetic field is not only responsible for an increase of hyperfine magnetic fields within the newly formed nanograins but also the corresponding components in the NFS time spectra are better identified via occurrence of quantum beats with higher frequencies. In order to distinguish between these two effects, simulated and experimental NFS time spectra obtained during in-situ temperature measurements with and without external magnetic field are compared.

## 1. Introduction

Iron-based metallic glasses (MG) exhibit excellent soft magnetic behaviour because of their high permeability and low coercivity [1,2]. Namely, these properties make them suitable candidates for increasing applications in industry [3] and helping to solve energy-saving problems [4]. Several studies of MGs including the crystallization process, their thermal and magnetic properties [5,6,7,8] were reported because they exhibit a wide range of useful physical and structural properties, especially from the application point of view [9,10]. Recently, powder prepared from crushed Fe${}_{78}$Si${}_{9}$B${}_{13}$ amorphous ribbons was used for production of transformer cores with improved magnetic properties obtained by suitable annealing [11] which was eventually performed also in an external magnetic field [12]. The effects of a magnetic field on structural transformations in MGs were reported earlier, for example, in [13]. Nevertheless, the studies performed so far have assessed the effects of an external magnetic field from the point of view of the resulting amount of the crystalline phase and/or their magnetic parameters that were reached after heat treatment. Here, we present a method that can monitor structural transformations in real-time, i.e., during the treatment, namely the so-called nuclear forward scattering (NFS) of synchrotron radiation.

This study aims at a thorough description of transformation processes in iron-based MGs during their conversion into nanocrystalline alloys. Understanding the mechanism of crystallization is important because of succeeding practical applications of these materials as well as from a point of view of basic physical phenomena that are related to structural transformations. To achieve this goal, we must apply novel analytical techniques. Well-established methods comprising X-ray diffraction and differential scanning calorimetry provide information that is averaged over the entire sample and are related exclusively to structural characterization. Recently, the method of NFS, which also scans the magnetic order of the studied systems via their hyperfine interactions, was applied to in-situ investigations of crystallization processes in Fe-Co-Mo-B-Cu MGs [14]. The influence of external magnetic field on the crystallization of FeZrB under isothermal conditions [15] and in FeCuMoB exposed to dynamical temperature increase [16] was studied too.

During crystallization, crystalline grains are formed within the amorphous matrix and, simultaneously, some elements are expelled to the grain boundary regions [15]. Thus, the grains differ from the amorphous matrix both in the long-range order arrangement and in composition. These quantities are reflected in hyperfine parameters which can be inspected by nuclear forward scattering. Formation and development of new crystalline phases in the material can be determined and followed by observing an appearance of new spectral components with different hyperfine parameters.

For example, in iron based amorphous alloys α-Fe nanocrystals develop in the amorphous matrix. Contrary to distribution of quadrupole splitting in the latter, crystalline grains are magnetically ordered, and their hyperfine magnetic fields are close to those of bulk α-Fe [17]. In addition, during the development of nanograins, their hyperfine parameters change too.

Nevertheless, there are cases when the NFS technique experiences difficulties in unambiguous identification of newly formed crystalline phases. This situation occurs when the nanograins are rather small in size (∼10 nm) and their magnetic moments exhibit thermal fluctuations, thus, resulting in an apparent collapse of the hyperfine splitting. In case of iron-based MGs, long-range order in crystalline phases is accompanied by magnetic ordering with rather strong hyperfine magnetic fields (>15 T). In this work, we discuss in detail crystallization processes that occur during the thermal treatment of two MGs with similar compositions, namely Fe${}_{57}$Co${}_{20}$Mo${}_{8}$Cu${}_{1}$B${}_{14}$ (in this MG Co atoms become part of bcc-Fe(Co) grains) and Fe${}_{75}$Mo${}_{8}$Cu${}_{1}$B${}_{16}$ in an external magnetic fields of 0.1 T and 0.652 T. Simulation of the influence of an external magnetic field on the corresponding NFS time spectra is also presented.

## 2. Experimental Details

Metallic glasses of Fe${}_{57}$Co${}_{20}$Mo${}_{8}$Cu${}_{1}$B${}_{14}$ and Fe${}_{75}$Mo${}_{8}$Cu${}_{1}$B${}_{16}$ compositions were prepared by planar-flow casting on a rotating quenching wheel. The as-prepared ribbons were 1–2 mm wide and ∼20 μm thick. To enhance the count rate, the samples were enriched with the isotope ${}^{57}$Fe to about 50% (note that the natural abundance of ${}^{57}$Fe is 2.17%).

NFS measurements were carried out at the nuclear resonance beamline of the European Synchrotron Radiation Facility (ESRF) in Grenoble. Photon beam energy of 14.413 keV and bandwidth of 3 meV was used to excite the ${}^{57}$Fe nuclei in the samples. Basic principles of the NFS technique are briefly described in [18], more details can be found for example in [19,20].

About 5 mm long pieces of ribbon-shaped samples were placed inside a vacuum furnace installed between two poles of an electromagnet and heated up to 600 ${}^{\circ }$C with a heating rate of 10 ${}^{\circ }$C/min. Experiments were performed without and with an external magnetic field of 0.1 T and 0.652 T. During annealing, each NFS time spectrum was accumulated for one minute. The incident linearly polarized beam entered the sample perpendicularly to its plane and the applied magnetic field was oriented parallel to the polarization axis. The experimental data obtained from NFS measurements were evaluated using the CONUSS software package (version 1.5 by W. Sturhahn, www.nrixs.com) [21,22] in combination with the sequential analysis tool Hubert [23]. During evaluation of the time spectra with the CONUSS software, we took into consideration the transversal coherence length of ∼10 μm [24,25], size of the nanograins <15 nm and thickness of the sample ∼20 μm (longitudinal coherence length extends far beyond the sample thickness). Under these assumptions, the photons scattered from structurally different regions of the metallic glass, viz. amorphous phase and nanograins, add up coherently.

## 3. Results and Discussion

### 3.1. Ferromagnetic Fe_57_Co_20_Mo_8_Cu_1_B_14_ Metallic Glass

Time spectra acquired from the NFS experiments performed upon the Fe${}_{57}$Co${}_{20}$Mo${}_{8}$Cu${}_{1}$B${}_{14}$ MG are shown in Figure 1. For the sake of more clear presentation of a high number of records, they are plotted as contour plots. The latter are stacked with respect to the duration time of the experiment which constitutes the vertical axes of the contour plots and is directly related to the temperature of annealing. The x-axes represent delayed time which has elapsed between the excitation pulse and the resonantly scattered photons. The counts of the registered photons (intensities) are colour coded in logarithmic scale.

Two well-distinguished transformations can be seen in Figure 1. The first one is situated at around 250 ${}^{\circ }$C and corresponds to ferromagnetic-to-paramagnetic transition at the Curie temperature. The studied MG is still amorphous and the corresponding beat patterns reflect qualitative change in the respective hyperfine interactions as demonstrated by selected NFS time spectra in Figure 2. Originally weak dipole magnetic interactions, which are characterized by quantum beats with high frequency (temperature 165 ${}^{\circ }$C, zero magnetic field), are in the paramagnetic state of the amorphous alloy replaced by electric quadrupole ones. The latter exhibit rather simple beat patterns, as, for example, those at 375 ${}^{\circ }$C.

The second qualitative change in the NFS time spectra can be seen between 395 ${}^{\circ }$C and 425 ${}^{\circ }$C. Reappearance of quantum beats featuring higher frequencies, which can be first noticed at 405 ${}^{\circ }$C in an external magnetic field of 0.652 T and even more enhanced at a higher temperature of 425 ${}^{\circ }$C, indicates growing importance of dipole magnetic interactions. They mark a presence of hyperfine magnetic fields which belong to ferromagnetic bcc-Fe(Co) nanocrystalline grains, i.e., the onset of crystallization. Chemical composition of this MG ensures that the onset of crystallization is very well visible because it is accompanied by a transition from a paramagnetic amorphous state to a strongly ferromagnetic crystalline state. It is noteworthy that the studied system is still amorphous and paramagnetic at 425 ${}^{\circ }$C when no external magnetic field is applied (Figure 2a).

For the analysis of the NFS time spectra, we have employed a fitting model consisting of five components: one distributed component for the amorphous phase and four components with hyperfine magnetic fields that were ascribed to the signals from bcc-Fe(Co) grains. The number of crystalline components was given by relative probabilities derived from binomial distribution corresponding to 0, 1, 2, and 3 Co atoms as nearest neighbours of Fe. Fitting parameters for the amorphous component comprised relative amount, quadrupole splitting and for all crystalline components relative amount, hyperfine magnetic field, and a parameter related to the effective thickness.

Temperature evolution of the relative amount of crystalline phase is shown in Figure 3. One can observe an apparent shift of the onset of crystallization towards lower temperatures with increasing external magnetic field. This confirms our assumption that the external magnetic field stimulates the nucleation of grains as it was reported also in other amorphous systems [15,16]. It is noteworthy that the onset of crystallization depends also on the strength of the external magnetic field which was oriented along the length of ribbon-shaped samples.

Hyperfine magnetic fields obtained from the fitting of the crystalline components of the NFS time spectra are shown in Figure 4. They start to appear after the onset of crystallization. With rising temperature of annealing, which also ensures an increase in the relative amount of the crystalline phase (see Figure 3), the hyperfine magnetic fields exhibit clear sharp values. With increasing temperature, the hyperfine magnetic field values follow the expected temperature dependence according to the Brillouin function.

### 3.2. Weak Magnetic Fe_75_Mo_8_Cu_1_B_16_ Metallic Glass

Unlike the previous case of well-established ferromagnetic Fe${}_{57}$Co${}_{20}$Mo${}_{8}$Cu${}_{1}$B${}_{14}$ MG where both magnetic and structural transformations are clearly visible from obvious deviations of the NFS beat patterns, the situation is quite different for a weak magnetic Fe${}_{75}$Mo${}_{8}$Cu${}_{1}$B${}_{16}$ MG. This composition is characterized by a close-to-room Curie temperature of the amorphous precursor ($T_{C}$ = 310 K), small size of bcc-Fe grains (<10 nm) and their low amount (<40%) even towards the end of the primary crystallization [26]. Thus, the onset of crystallization is not accompanied by notable changes in hyperfine interactions because due to its composition, this MG becomes paramagnetic already at the beginning of the heat treatment and the newly emerging Fe nanocrystals are too small and too scarce to ensure that their hyperfine magnetic fields will result in visible higher frequency quantum beats at the onset of crystallization.

Indeed, the acquired NFS time spectra exhibit simple beat patterns which maintain rather uniform structure over a broad temperature range as seen in the corresponding contour plots in Figure 5. The sample remains paramagnetic up to the final annealing temperature. However, as demonstrated by the results of diffraction of synchrotron radiation, this system starts to crystallise at the temperature of ∼450 ${}^{\circ }$C [14]. So, even though the bcc-Fe nanocrystals are formed, their presence cannot be confirmed via corresponding hyperfine magnetic fields, which are rather weak, and consequently, the NFS patterns do not show any remarkable changes in their shapes. That is why, alternative ways were proposed how to visualize the presence of nanograins by the help of other parameters derived from NFS time spectra [27].

Examples of experimentally obtained NFS time spectra (black symbols with error bars) together with the fitted curves (solid red lines) are presented in Figure 6 for selected temperatures of annealing. All NFS time spectra were evaluated using a two-component model consisting of one paramagnetic contribution and one component with weak magnetic interactions to refine the amorphous phase. Where necessary, additional narrow magnetic component was used to represent the newly formed crystalline phase. The fitted hyperfine parameters comprised relative amount of each component, quadrupole splitting and hyperfine field for weak magnetic component, hyperfine field for crystalline component and a parameter related to the effective thickness.

As already mentioned, all NFS patterns in Figure 5 and Figure 6 show very similar features. Even the application of weak external magnetic field of 0.1 T turned out to have not very pronounced effect and the NFS patterns were practically unchanged. Some deviations in the beat structure can be identified only towards higher temperatures of annealing (>500 ${}^{\circ }$C) in stronger external magnetic field of 0.652 T (Figure 5c and Figure 6c). Here, the newly formed nanocrystalline bcc-Fe grains can be identified via beat patterns with higher frequencies. We assume that the external magnetic field has contributed to their visibility by means of orientation of magnetic moments and consequent increase in magnetization. In this way they had stronger influence on the paramagnetic amorphous phase and, at the same time have formed more uniform magnetic structure of the crystalline phase with apparently stronger average hyperfine magnetic field. The latter is manifested via high-frequency quantum beats in the NFS patterns. Note that in the case of ferromagnetic Fe${}_{57}$Co${}_{20}$Mo${}_{8}$Cu${}_{1}$B${}_{14}$ MG, even more pronounced high frequency beats can be seen at much lower temperature (∼400 ${}^{\circ }$C) and in weak external magnetic field (Figure 2b). Presumably due to magnetic saturation of this soft magnetic metallic glass.

Temperature evolution of the relative amount of the crystalline magnetic phase and its hyperfine magnetic field are shown in Figure 7. They are presented only for the NFS experiment in an external magnetic field of 0.652 T as they were not visible in experiments performed in weaker magnetic fields. The amount of crystalline phase is rather low (<7 %) and the crystalline grains are quite small [26]. Consequently, their magnetic moments fluctuate especially at these high temperatures and are difficult to see through their hyperfine magnetic fields and the corresponding high frequency beat patterns. As far as the hyperfine field values of the crystalline phase are concerned, they follow the expected trend with increasing temperature as demonstrated in Figure 7b.

The obtained results suggest that an external magnetic field has an influence on the progress of crystallization especially when magnetic grains are formed. Nevertheless, a question still remains if the external magnetic field affects the process of crystallization as such or if it is only an effect of enhanced visualization of dipole magnetic hyperfine interactions in NFS time spectra especially in weak magnetic MGs. To decide which of these two assumptions is right, we performed simulations of NFS time spectra as presented below.

### 3.3. Simulations of the Impact of Magnetic Field on NFS Time Spectra

In this section, we describe estimation of the influence of an external magnetic field on a visibility of crystalline components in NFS experiments. Rapid increase of magnetization was reported when the investigated system was placed into an external magnetic field at a temperature close to its Curie temperature [28]. Because hyperfine magnetic field is proportional to magnetization, enhanced Zeeman splitting of nuclear levels occurs when the magnetization increases in the presence of external magnetic field. Consequently, we can observe changes in the shapes of NFS time spectra. Thus, in the Fe${}_{75}$Mo${}_{8}$Cu${}_{1}$B${}_{16}$ MG, where tiny nanograins are formed, the applied magnetic field triggers formation of magnetic quantum beats with higher frequency. Therefore, two effects can be involved, visualization of magnetic interactions caused by the applied magnetic field as mentioned above and direct influence of the magnetic field on the crystallization process itself.

To distinguish between these two cases, we performed simulations of NFS time spectra. In doing so, the following assumptions were made: (i) hyperfine magnetic field is proportional to the magnetization, (ii) the magnetization of nanocrystals depends on temperature, size of grains, and the applied magnetic field. In applying these assumptions, dependences for α-Fe will be considered including the evolution of Curie temperature with the mean grain size and the dependence of magnetization on temperature and external magnetic field [28,29]. Because in Fe-based MGs, which is also our case, a bcc-Fe crystalline phase is formed, this is a tolerable constraint. Particularly, because it is difficult to obtain values of magnetization from nanograins embedded in a residual amorphous matrix as the magnetic measurements provide integral information from the whole inspected volume.

Taking into account dependencies of magnetization on temperature (*T*), the applied magnetic field ($B_{\mathrm{ext}}$) and grain size (*d*) it is possible to construct a function $B_{\mathrm{hf}}\left(T,B_{\mathrm{ext}},d\right)$ which provides values of hyperfine magnetic field, $B_{\mathrm{hf}}$ for arbitrary temperature, applied magnetic field and mean grain size. Because no information on grains size is accessible from NFS experiments, we have used the following procedure. From evaluation of the experimental NFS time spectra for a selected temperature and/or external magnetic field, distribution of hyperfine magnetic fields is readily obtained. The corresponding distribution of grain sizes is calculated by an inverse function $d(T,B_{\mathrm{ext}},B_{\mathrm{hf}})$. From that function, we can calculate distribution of hyperfine magnetic fields for any other arbitrarily chosen temperature and/or magnetic field. Subsequently, hypothetical NFS time spectra can be simulated.

We demonstrate the above procedure for the NFS time spectrum that shows in Figure 8a high frequency quantum beats caused by the presence of crystalline grains at 577 ${}^{\circ }$C which have evolved in an external magnetic field of 0.652 T. Using the distribution of hyperfine fields obtained from the fitting, we have simulated NFS time spectrum for the same temperature how it would look like in zero magnetic field. The obtained simulated and the fitted curves are plotted in Figure 8 by blue and red curves, respectively, and compared with the experimental data. Figure 8b shows a situation where the experimental data with the corresponding fitted curve are presented for the same temperature but in zero external magnetic field and overlaid with the simulated spectrum.

It is noteworthy that the blue curves in Figure 8 represent a hypothetical NFS time spectrum as it would look like if the experiment were to be performed in zero magnetic field. It was simulated from the parameters that represent the higher frequency quantum beats from the in-field (0.652 T) experiment. Extrapolating the shapes of the NFS time spectra to zero-field conditions, notable deviations between the fitted experimental spectrum and the simulated one are observed in Figure 8b. This finding, however excludes the hypothesis that external magnetic field only affects the shapes of NFS time spectra thus making the higher frequency quantum beats better visible due to stronger hyperfine magnetic fields of the crystalline phase induced by enhanced magnetization. In other words, the high frequency quantum beats in the simulated NFS spectrum result from magnetic effects upon the formation of nanocrystalline grains. The same conclusion is supported by Figure 8a where deviations between the fitted and the simulated NFS spectrum can be seen, too.

## 4. Conclusions

Formation of nanocrystalline grains during in-situ heat treatment was followed by the help of nuclear forward scattering of synchrotron radiation. This method offers on fly information in real-time both on the structural arrangement and associated hyperfine interactions in the studied system. Possibilities of this unique technique were demonstrated using two types of Fe-based metallic glasses featuring opposite magnetic behaviour. Namely, we have used amorphous alloys with the compositions of Fe${}_{57}$Co${}_{20}$Mo${}_{8}$Cu${}_{1}$B${}_{14}$ and Fe${}_{75}$Mo${}_{8}$Cu${}_{1}$B${}_{16}$ which exhibit strong magnetic interactions and weak magnetic order, respectively.

Formation of ferromagnetic crystalline grains in the amorphous matrix is accompanied by occurrence of high frequency quantum beats in the NFS time spectra. They reflect the onset of crystallization differently in both investigated systems. While in ferromagnetic Fe${}_{57}$Co${}_{20}$Mo${}_{8}$Cu${}_{1}$B${}_{14}$ metallic glass the existence of bcc-Fe(Co) nanograins is readily seen, modifications of the shapes of NFS time spectra of a weakly magnetic Fe${}_{75}$Mo${}_{8}$Cu${}_{1}$B${}_{16}$ metallic glass are almost unnoticeable. All NFS experiments were performed also in an external magnetic fields of 0.1 T and 0.652 T.

For the Fe${}_{57}$Co${}_{20}$Mo${}_{8}$Cu${}_{1}$B${}_{14}$ metallic glass, a transition from ferromagnetic to paramagnetic state was observed at temperature of ∼250 ${}^{\circ }$C which was later followed by a structural transformation from amorphous to nanocrystalline arrangement. Evaluation of experimental NFS data revealed the influence of external magnetic field on the crystallization process. The applied magnetic field shifts the onset of crystallization towards lower temperatures. Simultaneously, it increases the fraction of crystalline grains embedded in the amorphous matrix.

In case of weak magnetic Fe${}_{75}$Mo${}_{8}$Cu${}_{1}$B${}_{16}$ metallic glass, presence of bcc-Fe nanograins during temperature annealing in zero field and in the weak external magnetic field of 0.1 T is not accompanied by obvious presence of corresponding quantum beats in the NFS time spectra. Therefore, identification of the onset of crystallization is not straightforward. This is mainly due to formation of tiny nanocrystalline grains whose magnetic moments significantly fluctuate especially at high enough temperatures. In the applied magnetic field of 0.652 T, we observed formation of nanocrystalline grains with sufficiently well-developed magnetic interactions.

To assess the influence of an external magnetic field upon the NFS experiments, we have performed simulations of the time spectra. The obtained results allow a conclusion that the accelerated crystallization, which was observed under the effect of external magnetic field in several metallic glasses investigated so far, is caused by magnetic energy from this field. The appearance of high frequency quantum beats in the shapes of NFS time spectra is not purely an artefact but a real demonstration of a presence of crystalline phase.

## Figures and Tables

**Figure 1 nanomaterials-09-00544-f001:**
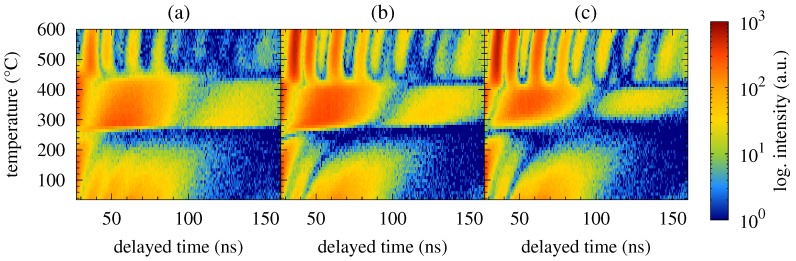
Contour plots of NFS time spectra recorded for the Fe${}_{57}$Co${}_{20}$Mo${}_{8}$Cu${}_{1}$B${}_{14}$ MG in an external magnetic fields of 0 T (**a**), 0.1 T (**b**) and 0.652 T (**c**).

**Figure 2 nanomaterials-09-00544-f002:**
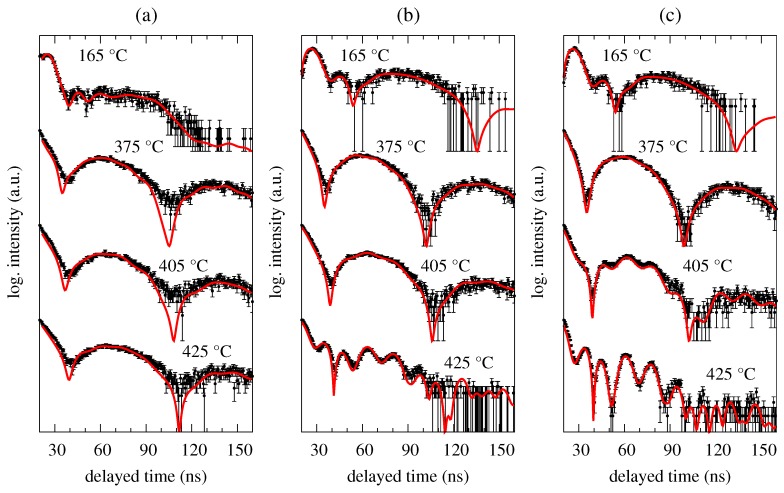
Selected NFS time spectra recorded for the Fe${}_{57}$Co${}_{20}$Mo${}_{8}$Cu${}_{1}$B${}_{14}$ MG in an external magnetic fields of 0 T (**a**), 0.1 T (**b**) and 0.652 T (**c**) at the indicated temperatures. Black symbols represent experimental data with error margins and red solid lines are results from their fitting.

**Figure 3 nanomaterials-09-00544-f003:**
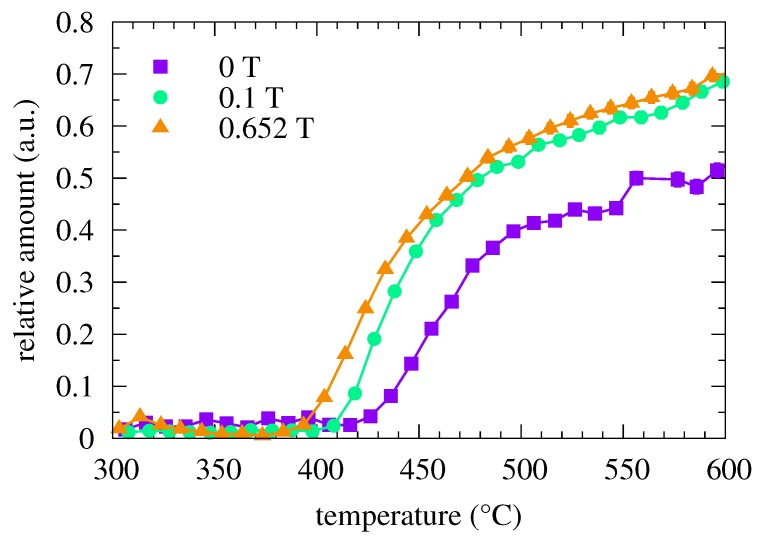
Relative amount of the crystalline phase in the Fe${}_{57}$Co${}_{20}$Mo${}_{8}$Cu${}_{1}$B${}_{14}$ MG plotted as a function of temperature for different values of external magnetic fields (see the legend).

**Figure 4 nanomaterials-09-00544-f004:**
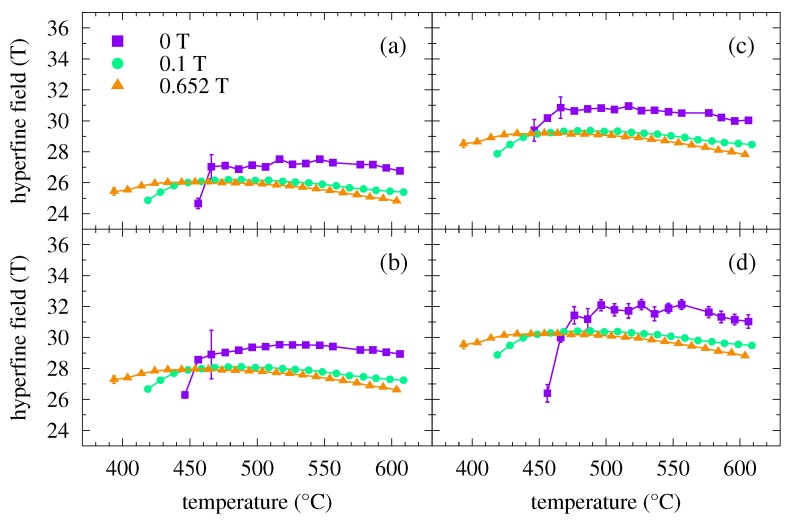
Hyperfine magnetic fields obtained from NFS time spectra of the Fe${}_{57}$Co${}_{20}$Mo${}_{8}$Cu${}_{1}$B${}_{14}$ MG plotted as a function of temperature for different values of external magnetic fields (see the legend). The plotted hyperfine magnetic fields correspond to the individual crystalline components which represent Fe atoms with 0 (**a**), 1 (**b**), 2 (**c**), and 3 (**d**) Co nearest neighbours.

**Figure 5 nanomaterials-09-00544-f005:**
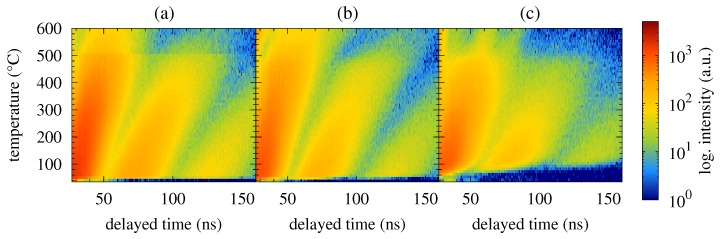
Contour plots of NFS time spectra recorded for the Fe${}_{75}$Mo${}_{8}$Cu${}_{1}$B${}_{16}$ MG in an external magnetic fields of 0 T (**a**), 0.1 T (**b**) and 0.652 T (**c**).

**Figure 6 nanomaterials-09-00544-f006:**
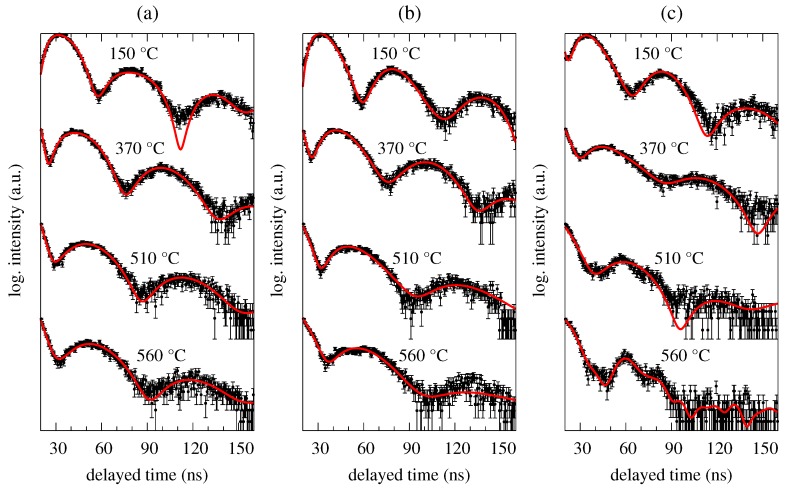
Selected NFS time spectra recorded for the Fe${}_{75}$Mo${}_{8}$Cu${}_{1}$B${}_{16}$ MG in an external magnetic fields of 0 T (**a**), 0.1 T (**b**) and 0.652 T (**c**) at the indicated temperatures. Black symbols represent experimental data with error margins and red solid lines are results from their fitting.

**Figure 7 nanomaterials-09-00544-f007:**
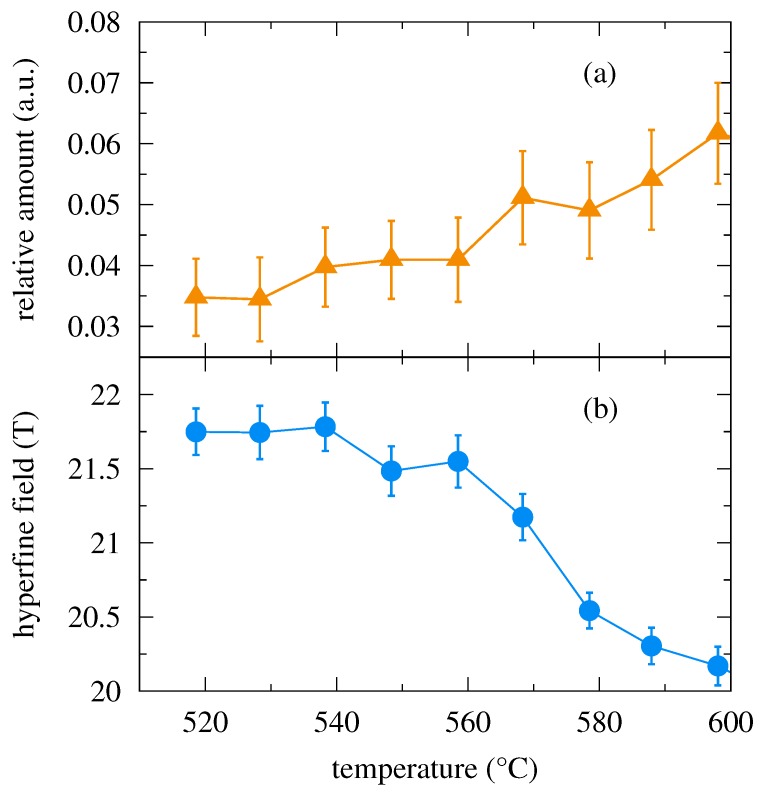
Relative amount of the crystalline phase (**a**) and the corresponding hyperfine magnetic field (**b**) as a function of temperature obtained from the evaluation of NFS time spectra of the Fe${}_{75}$Mo${}_{8}$Cu${}_{1}$B${}_{16}$ MG performed in an external magnetic field of 0.652 T.

**Figure 8 nanomaterials-09-00544-f008:**
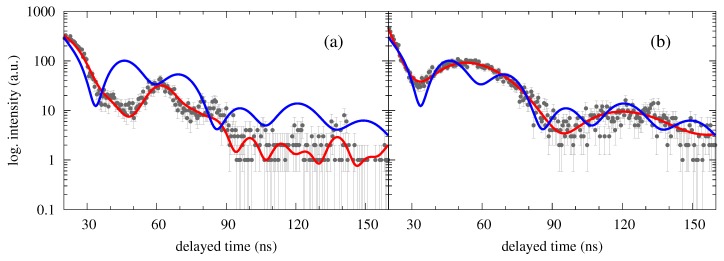
NFS time spectra of the Fe${}_{75}$Mo${}_{8}$Cu${}_{1}$B${}_{16}$ MG recorded at 577 ${}^{\circ }$C in an external magnetic field of 0.652 T (**a**) and in zero field (**b**). Red solid curves represent results from the fitting, blue curves are simulations for 0 T using the fitted data for 0.652 T (see text).

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
