# Peer review of "On the Formation of Nanocrystalline Grains in Metallic Glasses by Means of In-Situ Nuclear Forward Scattering of Synchrotron Radiation"

_nanomaterials, 2019, doi:10.3390/nano9040544_

Reviewer 1 Report

The authors of the manuscript present exciting novel results with a unique technique developed by their group previously. The method is very efficient and offers the only feasible approach to follow the formation of nanocrystalline grains from amorphous metallic glasses and their hyperfine interactions in real time. The novelty of the presented study is the detailed analysis of the effect of an external magnetic field on the grain formation process for two different metallic glasses in a broad temperature range. The presentation is sound, the figures are clear. Therefore, the paper definitely deserves to be published in Nanomaterials.

Nevertheless, I have a few comments concerning methodological clearness and text editing.

In Line 37, the composition Fe-Co-M-B-Cu should be changed to Fe-Co-Mo-B-Cu.

In Lines 52–54, the sentence “This situation occurs when the nanograins are rather small in size (~10 nm) and their magnetic moments exhibit thermal fluctuations, thus resulting in weak magnetic interactions.” In fact, thermal fluctuations lead to relaxation phenomena that result in an apparent collapse of the hyperfine splitting, however, strictly speaking, this can hardly be called a “weak magnetic interaction”.

It is not quite clear, how the fitted model functions, i.e. the red curves in Figs. 2, 6 and 8 were calculated. Indeed, the value of the transverse coherence length of the photons, a function of the geometry of the experiment (cf. A.Q.R. Baron et al., Phys. Rev. Lett. 77, 4808 (1996)) has not been specified. Depending on whether the average transverse distance of the amorphous regions and the nanocrystalline grains is greater or less than the transverse coherence length, their contributions to the quantum beats should be added incoherently or coherently, respectively (cf. H.F. Grünsteudel et al., Hyperfine Interact. C 1, 509 (1996).). As I believe, the authors may wish to clarify this point in the manuscript.

I suggest that, after these minor changes, the manuscript can be published in Nanomaterials.

Author Response

The authors thank the reviewer for his/her effort with evaluating the manuscript.

Ad 1: We replaced Fe-Co-M-B-Cu by Fe-Co-Mo-B-Cu.  This was our mistake. We are grateful to the reviewer for his/her precise evaluation/reading of our manuscript.

Ad 2: The reviewer has sound arguments and we have changed the end of the sentence at line 54 and omitted “weak magnetic interactions”.

Ad 3: The transverse coherence (A.Q.R. Baron et al., Phys. Rev. Lett. 77, 4808 (1996), DOI:10.1103/PhysRevLett.77.4808. and A.Q.R. Baron, Hyperfine Int. 123, 667-680 (1999), DOI:10.1023/A:1017065100051) is typically ~10 μm. We suppose rather homogeneous distributions of the nanograins in the whole volume (some minor deviations could occur on the surfaces). The mean size of nanograins is much lower than the transverse coherence. So, all structurally different regions in the sample, viz. amorphous phase and nanocrystallites, are present inside the volume of coherence (x-coh*y-coh*sample thickness). Consequently, we can consider that the photons are added coherently.

We also addressed this point in the manuscript by adding new text in lines 76 – 80.

Reviewer 2 Report

Here authors present their research of metallic glass crystallization via nuclear forward scattering method. The results and experiment are interesting. The paper well written and I agree with most of it. However some points should be clarified before publication.

1.  String N18 in PDF file - Author talk about several studies but provide only one reference here. Please add more.

2. Str. N37. Please add an explanation what is M in Fe-Co-M-B-Cu.

3. Notation in composition formula (Fe2.85Co1)77Mo8Cu1B14 is confusing. Please provide amount of Fe and Co in unit fraction.

4. How such alloy compositions were chosen? What is their importance? Are these new compositions or well known ones? If these are new please provide more information on their crystallization (Tg, Tx, crystallization phases, may be DSC curves). If not please provide references. For now your statements about size of the grains are not supported by any direct evidence. It would be good to see TEM images of these crystals.

5. Please give explanation of the red lines in Fig. 2 and 6 in the caption.

6. Please add to the experimental part the information about field orientation during annealing.

Author Response

The authors thank the reviewer for his/her effort with evaluating the manuscript.

Ad 1: We added another 5 references related to crystallization process, magnetic and structural properties and applications.

The modified sentence is at lines 17 - 20:

Several studies of MGs including the crystallization process, their thermal and magnetic properties [Zhao2019, Parsons2017, Salazar2017, Barandiaran2011] were reported because they exhibit a wide range of useful physical and structural properties, especially from the application point of view [Gutierrez2017, 5].”

Zhao2019 DOI:10.1016/j.jmmm.2018.08.058

Parsons2017 DOI:10.1016/j.jallcom.2017.06.208

Salazar2017 DOI:10.1088/1361-6463/50/1/015305

Barandiaran2011 DOI:10.1002/pssa.201000738

Gutierrez2017 DOI:10.3390/s17061251

Ad 2: This was our mistake. We corrected it, M was replaced by Mo (Molybdenum) to Fe-Co-Mo-B-Cu. We are grateful to the reviewer for his/her precise evaluation/reading of our manuscript.

Ad 3: We have changed the composition of (Fe2.85Co1)77Mo8Cu1B14 to Fe57Co20Mo8Cu1B14 and made necessary modifications in the manuscript.

Ad 4: The current compositions of metallic glasses were chosen to demonstrate diversities in the temperature evolution of the NFS time spectra that were recorded from metallic glasses with different magnetic behaviour but of (principally) the same origin. Note that they were obtained from the same precursor with a nominal composition of (Fe1-xCox)76Mo8Cu1B15 (0 ≤ x ≤ 0.75) which we have already studied some time ago (see Miglierini M.: Hyperfine Interactions and Crystallization Kinetics of Co-Substituted NANOPERM-Type Alloys, J. Phys.: Conf. Series 217 (2010) 012092, DOI: 10.1088/1742-6596/217/1/012092). For the sake of NFS experiments we have prepared samples enriched in 57Fe. A thorough check of their composition by chemical methods have shown small deviations with respect to the intended nominal composition. This is the reason why we have written the composition as pointed by the reviewer in his/her comment Ad 3. Nevertheless, in this manuscript this composition was modified as stated in our reply Ad 3.

More information about both systems can be found in our previous papers [14, 26] that are listed among the current references. Namely, TEM images and DSC of the Fe-Mo-Cu-B alloy are shown in Fig. 3 and Fig. 1, respectively, in [26]. DSC of the Fe-Co-Mo-Cu-B metallic glass is shown in Fig. 8 in [14]. Crystallization temperatures for both systems are provided in Fig. 5 in [14].

We believe that the readers interested in more information on the current metallic glasses can find relevant data in the provided references. In order not to extend the length of the manuscript far beyond the set limits, we have not discussed the details on the crystallites. In addition, the main message of this contribution is aimed at pointing the possibilities of NFS for in-situ study of structural transformations in metallic glasses in rather general terms.

Ad 5: We added an explanation of the red lines to the captions of figures 2 and 6. It reads:

Black symbols represent experimental data with error margins and red solid lines are results from their fitting.”

Ad 6: We added  an explanation of the direction of the magnetic field at lines 72 – 74 and more about  experimental setup at lines 69 – 70

Nanomaterials EISSN 2079-4991 Published by MDPI AG, Basel, Switzerland RSS E-Mail Table of Contents Alert
Back to Top